# Green Synthesis of Silver Oxide Nanoparticles from *Mauritia flexuosa* Fruit Extract: Characterization and Bioactivity Assessment

**DOI:** 10.3390/nano14231875

**Published:** 2024-11-22

**Authors:** Johana Zúñiga-Miranda, David Vaca-Vega, Karla Vizuete, Saskya E. Carrera-Pacheco, Rebeca Gonzalez-Pastor, Jorge Heredia-Moya, Arianna Mayorga-Ramos, Carlos Barba-Ostria, Elena Coyago-Cruz, Alexis Debut, Linda P. Guamán

**Affiliations:** 1Centro de Investigación Biomédica (CENBIO), Facultad de Ciencias de la Salud Eugenio Espejo, Universidad UTE, Quito 170527, Ecuador; johana.zuniga@ute.edu.ec (J.Z.-M.); dsvaca2@gmail.com (D.V.-V.); saskya.carrera@ute.edu.ec (S.E.C.-P.); rebeca.gonzalez@ute.edu.ec (R.G.-P.); jorgeh.heredia@ute.edu.ec (J.H.-M.); arianna.mayorga@ute.edu.ec (A.M.-R.); 2Centro de Nanociencia y Nanotecnología, Universidad de Las Fuerzas Armadas ESPE, Sangolquí 171103, Ecuador; ksvizuete@espe.edu.ec (K.V.); apdebut@espe.edu.ec (A.D.); 3Escuela de Medicina, Colegio de Ciencias de la Salud Quito, Universidad San Francisco de Quito USFQ, Quito 170901, Ecuador; cbarbao@usfq.edu.ec; 4Instituto de Microbiología, Universidad San Francisco de Quito USFQ, Quito 170901, Ecuador; 5Carrera de Ingeniería en Biotecnología, Universidad Politécnica Salesiana, Quito 170143, Ecuador; ecoyagoc@ups.edu.ec; 6Departamento de Ciencias de la Vida y Agricultura, Universidad de las Fuerzas Armadas ESPE, Sangolquí 171103, Ecuador

**Keywords:** silver nanoparticles, *Mauritia flexuosa*, biological activity, green synthesis, anticancer, multidrug-resistant bacteria, antibacterial agent, sustainable approach

## Abstract

The increasing prevalence of multidrug-resistant (MDR) pathogens, persistent biofilms, oxidative stress, and cancerous cell proliferation poses significant challenges in healthcare and environmental settings, highlighting the urgent need for innovative and sustainable therapeutic solutions. The exploration of nanotechnology, particularly the use of green-synthesized nanoparticles, offers a promising avenue to address these complex biological challenges due to their multifunctional properties and biocompatibility. Utilizing a green synthesis approach, *Mauritia flexuosa* Mf-Ag_2_ONPs were synthesized and characterized using dynamic light scattering (DLS), transmission electron microscopy (TEM), X-ray diffraction (XRD), energy-dispersive X-ray spectroscopy coupled with scanning electron microscopy (EDS-SEM), UV-Vis spectroscopy, and Fourier transform infrared spectroscopy (FTIR). The Mf-Ag_2_ONPs exhibited potent antibacterial effects against both non-resistant and MDR bacterial strains, with minimum inhibitory concentrations (MICs) ranging from 11.25 to 45 µg/mL. Mf-Ag_2_ONPs also demonstrated significant antifungal efficacy, particularly against *Candida glabrata*, with an MIC of 5.63 µg/mL. Moreover, the nanoparticles showed strong biofilm inhibition capabilities and substantial antioxidant properties, underscoring their potential to combat oxidative stress. Additionally, Mf-Ag_2_ONPs exhibited pronounced anticancer properties against various cancer cell lines, displaying low IC_50_ values across various cancer cell lines while maintaining minimal hemolytic activity at therapeutic concentrations. These findings suggest that Mf-Ag_2_ONPs synthesized via an eco-friendly approach offer a promising alternative for biomedical applications, including antimicrobial, antifungal, antioxidant, and anticancer therapies, warranting further in vivo studies to fully exploit their therapeutic potential.

## 1. Introduction

Antimicrobial compounds like antibiotics have been crucial in treating infections; however, their misuse across humans, animals, and agriculture has accelerated the emergence of MDR bacteria, which now pose a major global health threat, causing an estimated 4.95 million deaths in 2019 [1,2,3]. Alongside antimicrobial resistance, persistent biofilms, oxidative stress from free radicals, and cancer remain significant health challenges, requiring innovative strategies [4,5,6]. To address these challenges, nanotechnology has emerged as an innovative solution. This field specializes in the ability to observe, measure, manipulate, assemble, control, and manufacture matter at the nanometer scale (1–100 nm) [7]. Nanoparticles have been used in various medical fields due to their controlled particle size and physicochemical properties, which depend on their electronic, electrical, mechanical, magnetic, thermal, dielectric, optical, and biological behaviors [8,9]. Nanoparticles are nanosized particles with a large surface area that enables them to penetrate bacterial cells, making them a promising alternative to conventional antimicrobial agents [9].

Nanoparticles are synthesized using two main methods: top-down and bottom-up. Top-down methods break larger molecules into high-energy, -temperature, or -pressure nanoparticles, requiring expensive equipment [10,11,12]. Bottom-up methods use chemical or biological processes to stabilize metallic atoms from simpler substances. Chemical methods involve controlled reactions with metallic precursors and reducing agents at the nanoscale, but can produce hazardous waste, prompting interest in eco-friendly biological approaches [10,12]. Green synthesis of nanoparticles provides eco-friendly and sustainable alternatives to traditional chemical synthesis. It reduces the use of toxic chemicals, reduces costs, and minimizes environmental impact while often utilizing renewable resources. Plant-based synthesis (phytosynthesis) is particularly favored due to reduced processing time, toxicity, and energy consumption, with the resulting nanoparticles being environmentally friendly [13,14]. The study of nanoparticles has increased in recent years, with metals like Au, Ag, Cu, Pt, and Zn being used in green synthesis from aqueous plant extracts for a range of medical and pharmaceutical applications.

Phytosynthesis of nanoparticles, particularly silver oxide nanoparticles (Ag_2_ONPs) using plant extracts, is an emerging green chemistry approach that leverages natural plant compounds to reduce and stabilize metal ions, forming nanoparticles [15,16]. Ag_2_ONPs have exhibited antioxidant, anticancer, antifungal, anti-inflammatory, and wound-healing properties. Phytomolecules such as flavonoids, alkaloids, terpenoids, and other natural compounds in plant extracts play a critical role in reducing silver ions and capping and stabilizing Ag_2_ONPs [17]. These biomolecules facilitate the reduction process and contribute to the stability and unique properties of biosynthesized nanoparticles [18]. Moreover, plant metabolites like flavonoids and terpenoids from plants such as geranium leaves serve as feasible reducing substances for silver ions, further contributing to the green synthesis of metallic nanoparticles [18].

*M. flexuosa*, commonly known as “aguaje”, is rich in lipids and carotenoids, particularly β-carotene and tocopherols, which are valued for their nutritional and medicinal properties, including anti-inflammatory and antioxidant effects. It is also a significant source of vitamins, such as ascorbic acid, and contains essential fatty acids and antioxidants that interact with free radicals and reactive oxygen species, protecting cells and tissues from oxidative damage [19,20]. Additionally, the fruit has been traditionally used to heal and protect the skin from damage. It is often utilized in the cosmetic industry for products such as sun creams and treatments for burns and skin aging due to its cicatrizing and anti-aging properties [21,22].

In this study, silver oxide nanoparticles using *M. flexuosa* fruit extract were synthesized and their biological activities were assessed. The efficacy of these nanoparticles was evaluated through their antimicrobial activity, including MDR strains, their potential to inhibit biofilm formation, and their antioxidant and hemolytic capacities. Moreover, we investigated their impact on different cancer cell lines to determine their potential anticancer effects. By taking this comprehensive approach, we aim to maximize the advantages of using *M. flexuosa* fruit extract in the phytosynthesis of silver oxide nanoparticles, thereby providing valuable insights into their potential medical and environmental applications.

## 2. Materials and Methods

### 2.1. Preparation of M. flexuosa Fruit Extract and Phytosynthesis of Mf-Ag_2_ONPs

The fruit pulp powder was acquired from the commercial brand Morevit^TM^ produced by the dehydration process of the ripe fruits of *M. flexuosa*. To make the aqueous extract, a 1:17 (*w*/*v*) ratio was used and it was mixed over a magnetic stirrer hot plate at 60 °C for 10 min. The resultant extract was filtered through Whatman filter paper No. 1 and a polyethersulfone membrane filter 0.45 µm. The extract was dried in the freeze-dryer, obtaining an extraction percentage of 19.5%. The dry extract was stored at 4 °C for further use.

Aqueous extract of *M. flexuosa* (20 mL) was added to 20 mL of 5 mM silver nitrate solution. The reaction was carried out on a hot plate at 60 °C for 90 min with continuous stirring. A change in the color of the solution indicated the formation of Mf-Ag_2_ONPs. After that, the suspension was centrifuged at 13,000× *g* for 10 min. The resulting sediment was washed three times with distilled water. The pure Mf-Ag_2_ONPs were freeze-dried and stored at 4 °C for further characterization and biological studies.

### 2.2. Determination of Total Polyphenol Contents

A gallic acid stock solution was prepared in water at a 1.0 mg/mL concentration. The solution was diluted to 400 µg/mL, and eight standard solutions were prepared by serial two-fold dilutions. Calibration curves were generated using these standard solutions, and water was used as a blank.

The extract’s total polyphenol content was determined using the Folin–Ciocalteu assay, as described by Johnson et al. [23]. In a 96-well microplate, 100 µL of a 1:10 diluted Folin–Ciocalteu reagent was mixed with 20 µL of extract or standard solution. The mixture was gently mixed before being incubated in the dark at room temperature for 10 min. Next, 100 µL of 7.5% aqueous sodium carbonate solution was added. After gentle mixing, the microplate was incubated in the dark at room temperature for 30 min. The absorbance was then measured at 750 nm using a Cytation 5 (BioTek) microplate reader after 300 s of linear agitation. The data are presented as a mean ± standard deviation of a triplicate experiment, expressed in mg of gallic acid equivalent GAE/100 g of dry extract.

### 2.3. Identification of Phenolic Compounds

Phenolic compounds were quantified as described by Coyago-Cruz et al. (2023) [24]. The resulting aqueous extract was passed through a 0.45 µm PVDF filter and placed in 2 mL vials for further analysis. Phenolics were separated and quantified using an RRLC 1200 liquid chromatograph (Agilent Technologies, Santa Clara, CA, USA) with a DAD-UV-VIS detector operating in the wavelength range of 220 to 500 nm. A Zorbax Eclipse Plus C18 column (4.6 × 150 mm, 5 microns) from Agilent Technologies (Santa Clara, CA, USA) was employed. The mobile phase corresponded to an aqueous solution with 0.01% formic acid (solvent A) and pure acetonitrile (solvent B). A flow rate of 1 mL/min was set, and a linear elution gradient was applied, starting with 100% A at 0 min, transitioning to 95% A + 5% B at 5 min and 50% A + 50% B at 20 min, with a column wash and re-equilibration at 22 min. Each sample injection used a volume of 10 µL, with two injections per extract. Analysis of resulting chromatograms was conducted using OpenLab ChemStation software (Version 2.15.26), where phenolic compounds were identified by comparing retention times and UV-Vis spectra at specific wavelengths like 280, 320, and 370 nm. Quantification of phenolics relied on a calibration curve with 1 mg/mL standards of various compounds like caffeic acid, chlorogenic acid, chrysin, *p*-coumaric acid, *m*-coumaric acid, *o*-coumaric acid, ferulic acid, gallic acid, *p*-hydroxybenzoic acid, 3-hydroxybenzoic acid, 2,5-dihydroxybenzoic acid, kampherol, luteolin, naringin, quercetin, rutin, shikimic acid, syringic acid, and vanillic acid. These standards were individually prepared and quantified with 3, 5, 10, 15, and 20 µL injection volumes. The concentration of phenolics was expressed as milligrams per 100 g dry weight (mg/100 g DW).

### 2.4. Characterization of Mf-Ag_2_ONPs

The formation of nanoparticles was confirmed by the appearance of a characteristic absorption peak, using UV-Vis spectrophotometer (Cytation5 multi-mode detection system, BioTek, Agilent Technologies, Santa Clara, CA, USA), scanning across a wavelength range of 250 to 750 nm.

The hydrodynamic size distribution was analyzed using a dynamic light-scattering (DLS) instrument (LB-550, Horiba, Kyoto, Japan) under the following analysis parameters: 25 °C, viscosity 0.9190 mPa/s, host medium refractive index 0.330, and silver refractive index 0.280. ζ- was measured using a Horiba nanoPartica SZ-100 nanoparticle analyzer.

Samples were characterized by X-ray diffraction (XRD) to evaluate the crystalline structure of the particles. For this purpose, a Malvern Panalytical Empyrean X-ray diffractometer equipped with a copper X-ray tube (Kα radiation, λ = 1.54056 Å) was used. The XRD data were collected in the 2Θ range from 10° to 90° at 45 kV and 40 mA. Highscore© software (Version 4.9a (4.9.1.29739), Malvern Panalytical B.V., Almelo, The Netherlands) was used for data interpretation with the Crystallography Open Database (COD). 

The morphology and size of the nanoparticle synthesis were observed at 80kV by transmission electron microscopy (TEM) in a FEI-Tecnai G20 Spirit Twin transmission electron microscope equipped with an Eagle 4 k HR camera and an LaB6 filament.

Energy-dispersive X-ray spectrometry (EDS) and elemental mapping were performed in a field emission scanning electron microscope (Mira 3, Tescan, Brno, Czech Republic) in order to evaluate the semiquantitative elemental composition of the obtained nanoparticles. The EDS detector used for this analysis was the X-Flash 6|30 (Bruker, Billerica, MA, USA), which has a resolution of 123 eV at Mn Kα.

FTIR measurements were used to determine the presence of distinct functional groups in biomolecules participating in Ag_2_ONP generation and nanoparticle capping/stabilization. FTIR spectra were recorded by a Perkin Elmer FTIR Spectrum One by using an ATR system (4000–650 cm^−1^)

### 2.5. Antimicrobial Activity Assay

The antibacterial activity of the aqueous extract of *M. flexuosa* and biosynthesized Mf-Ag_2_ONPs was assessed against four non-MDR bacteria: *Escherichia coli* ATCC 25922, *Staphylococcus aureus* ATCC 25923, *Pseudomonas aeruginosa* ATCC 27853, and *Burkholderia cepacia* ATCC 25416. All strains were obtained from the American Type Culture Collection (ATCC, Manassas, VA, USA) and maintained at −80 °C with 25% (*v*/*v*) glycerol supplementation. Additionally, MDR bacteria *Klebsiella pneumoniae*, *E. coli*, *Salmonella enterica* serovar Kentucky, *Enterococcus faecium*, and *P. aeruginosa* were tested. These five clinical multidrug-resistant isolates were provided by the National Health Institute of Ecuador (INSPI).

The minimal inhibitory concentration (MIC) was determined using the microdilution method according to the Clinical and Laboratory Standards Institute (CSLI) guidelines [25]. This assay was performed in a 96-well microtiter plate. The Mf-Ag_2_ONPs were serially diluted in distilled water; then, 10 μL of each dilution was added to 190 μL of bacterial suspension (5 × 10^5^ CFU/mL) to a total volume of 200 μL. The final concentration of Mf-Ag_2_ONPs in each well ranged from 1.40 to 90 μg/mL. The plates were then incubated at 37 °C for 24 h with constant shaking at 300 cpm (double orbital setting), and the OD_600_ was monitored in a Cytation5 multi-mode detection system (BioTek). The MIC was determined by comparing the OD_600_ at time 0 with the value after 24 h in samples exposed to the tested suspension at different concentrations as well as positive and negative controls of inhibition.

Ampicillin (100 µg/mL) and nourseothricine (100 µg/mL) served as the positive controls to non-MDR and MDR bacteria, respectively. The negative control consisted of the bacteria suspension alone (without Mf-Ag_2_ONPs). The MIC was defined as the lowest concentration of the antibacterial agent, which completely inhibited the growth of the microorganism as determined by the optical density at 600 nm. These assays were performed at least in triplicate. 

### 2.6. Antifungal Activity Assay

The MIC of Mf-Ag_2_ONPs against four Candida strains (*Candida krusei* ATCC 14243, *Candida albicans* ATCC 10231, *Candida glabrata* ATCC 66032, and *Candida tropicalis* ATCC 13803) was determined using the microdilution method according to the CSLI guidelines with the following modifications: Mf-Ag_2_ONPs were serially diluted in distilled water (1.25–100 μg/mL); then, 10 μL of each dilution was added to 190 μL of fungal suspension (5 × 10^5^ cfu/mL) to a total volume of 200 μL. The plates were then incubated at 37 °C for 72 h with constant shaking (200 rpm). The OD_600_ was determined immediately after inoculation (time 0) and at the end of the 72h incubation in a Cytation5 multi-mode plate reader (BioTek). The MIC was determined by comparing the OD_600_ at time 0 with value after 72 h in samples exposed to the tested suspension at different concentrations as well as positive (nourseothricin, 100 µg/mL) and negative controls (H_2_O). The MIC was defined as the lowest concentration of the Mf-Ag_2_ONPs, which completely inhibited the growth of the microorganism as determined by the optical density at 600 nm. These assays were performed at least in triplicate. 

### 2.7. Biofilm Inhibition Activity

*S. aureus* ATCC 25923, *L. monocytogenes* ATCC 13932, *P. aeruginosa* ATCC 9027, and *B. cepacia* ATCC 25416 were assessed for biofilm inhibition. Biofilms of ATCC strains were cultured in TSB+G (tryptic soybean medium supplemented with 1% glucose) overnight at 37 °C.

The next day, overnight cultures were diluted 1:100 in TSB+G and aliquoted in volumes of 150 µL into 96-well plates. Mf-Ag_2_ONPs were prepared in distilled water and distributed in concentrations ranging from 2.5 to 40 µg/mL into each respective well. Plates were incubated under static conditions at 37 °C for 24 h.

Concentrations were tested in technical replicates during three independent biological replicates.

After incubation, the medium was carefully removed with a micropipette. Subsequently, each 96-well plate was washed two times with PBS buffer 1x (pH 7.2) and dried inside a laboratory oven at 60 °C for 1 h. After the fixation step, 150 µL of 0.1% crystal violet solution was added to each well and incubated at room temperature on the countertop for 20 min. The staining solution was discarded, wells were washed thrice with PBS buffer 1x (pH 7.2), and 150 µL of 98% ethanol was carefully added to each well and incubated for 30 min (room temperature). Finally, absorbance was measured at 570 nm using the Cytation5 multi-mode plate reader (BioTek). The following formula was used to determine the biofilm inhibition percentage: $$Inhibitory\;rate\;\left(\%\right)=100*\frac{OD570\mathrm{nm}\;\left(Positive\;control\right)-OD570\mathrm{nm}\;\left(Sample\right)}{OD570\mathrm{nm}\;\left(Positive\;control\right)}$$

### 2.8. Antioxidant Activity

For this purpose, the DPPH (2,2-diphenyl-1-picrylhydrazyl) radical scavenging assay and the ABTS decolorization assay were carried out using nanoparticles in water and the extract alone. The stock solutions for the nanoparticles were prepared by dissolving them in water to a final concentration of 50 µg/mL, whereas the extract was adjusted to a final concentration of 2.34 mg/mL in water. Ascorbic acid was used as an antioxidant standard, and the stock was prepared in water to 200 µg/mL. 

The *DPPH* assay was adapted from the literature [26] and performed in 96-well plates. In brief, the stock solutions of the nanoparticles, the extract, and the standard were prepared as serial two-fold dilutions in methanol to a final volume of 100 µL. Then, 100 µL of a 0.2 mM methanol solution of *DPPH* radicals was added to 100 µL of methanolic solution. Finally, the mixture was incubated in the dark at room temperature for 40 min, and the absorbance was read at 515 nm on a Cytation5 multi-mode plate reader (BioTek). The following formula was used to calculate the % *DPPH scavenging* activity:$$\%\;DPPH\;scavenging=100*\;\frac{A_{sample+DPPH}-A_{sample\;blank}}{A_{DPPH}-A_{solvent}}$$

The ABTS decolorization assay was adapted from the literature [27]. Briefly, a 7 mM ABTS solution was mixed with 245 mM ammonium persulfate (APS) solution to a final concentration of 2.45 mM APS. The solution was incubated in the dark for 16 h and then diluted in water until the absorbance was ~0.7 at 734 nm (ABTS radical solution). A 2 mM Trolox stock was prepared in 1x PBS pH 7.4, and then it was serially diluted to obtain solutions with concentrations ranging from 12.5 to 400 µM to generate a calibration curve to determine the samples’ Trolox equivalent (TE) antioxidant capacity (TEAC) values. For this purpose, the concentrations of Trolox that produced the same percent reduction in absorbance at 734 nm for the nanoparticles, the extract, and the ascorbic acid were calculated. These values were expressed as µmol TE/g. 

Additionally, the stock solutions of the nanoparticles, the extract, and the ascorbic acid were prepared as serial two-fold dilutions in water (solvent). Then, in a 96-well microtiter plate, 10 µL of each dilution was mixed with 190 µL of ABTS radical solution (~0.7 at 734 nm). Finally, the mixture was incubated in the dark at room temperature for 5 min, and the absorbance was read at 734 nm on a Cytation 5 (BioTek) plate reader.

The following formulas were used to calculate the % decolorization:$$\%\;Decolorization=100*\;\frac{A_{solvent}-A_{sample\;}}{A_{solvent}}$$

The antioxidant capacity relative to *Trolox* was calculated using the equation obtained from the calibration curve and the formula:$$Trolox-eq\;\left(µ\mathrm{mol}/g\right)=\frac{Sample\;decolorization\;\left(\%\right)-b}{a}/Sample\;concentration\;\left(g/L\right)$$

IC_50_ values were used to express the antioxidant activity of each compound and represented the concentration that could scavenge 50% of the DPPH free radical or decolorize 50% of the ABTS radical solution. These values were determined by using GraphPad Prism 10.2 (GraphPad Software Corp, San Diego, CA, USA). All the results are given as a mean ± standard deviation (SD) of experiments performed at least in triplicate.

### 2.9. Anticancer Activity

The study used HeLa (human cervical carcinoma, ATCC No. CCL-2), MDA-MB-231 (human breast adenocarcinoma, ATCC No. HTB-26), HCT116 (human colorectal carcinoma, ATCC No. CCL-247), HT29 (human colorectal adenocarcinoma, ATCC No. HTB-38), and NIH3T3 (mouse NIH/Swiss embryo fibroblasts, ATCC No. CRL-1658) obtained from ATCC. THJ29T (human thyroid carcinoma, Cat. No. T8254) was obtained from Applied Biological Materials Inc. (abm, Richmond, BC, Canada) and is also documented in the relevant literature [28]. Cells were cultured at 37 °C, 5% CO_2_ in Dulbecco Modified Eagle’s medium/F-12 (DMEM/F-12) (Sigma-Aldrich, St Louis, MO, USA), supplemented with 10% Fetal Bovine Serum (FBS) (Eurobio, Les Ulis, France) and 1% penicillin and streptomycin (Thermo Fisher Scientific, Gibco, Miami, FL, USA). To assess the impact of the compounds on cell proliferation, cells were seeded in 96-well plates at a density of 1 × 10^4^ cells/well. Mf-Ag_2_ONPs were added to the wells at final concentrations ranging from 0.39 to 250 μg/mL and incubated with the cells for 72 h. Mf extract was also incubated with the cells from 30 to 4000 μg/mL. Following incubation, the MTT (thiazolyl blue tetrazolium bromide) dye assay was conducted according to the manufacturer’s instructions. Briefly, 10 μL of MTT solution (5 mg/mL) was added to each well, and the plates were incubated for 1–2 h. After incubation, the media were removed, and 50 μL of DMSO was added to each well to dissolve the formazan crystals. The plate was gently agitated for 5 min to ensure homogeneous dissolution, and the absorbance was measured at 590 nm using a Cytation5 multi-mode detection system (Biotek). Dose–response curves were generated using untreated cells as the 100% cell proliferation control to determine the IC_50_ (concentration of compound inhibiting 50% of cell proliferation). Values are expressed as mean ± standard deviation, n = 4. The therapeutic index (TI) was calculated as the ratio between IC_50_ (non-tumor cells, NIH3T3) and IC_50_ (tumor cells). GraphPad Prism 10.2 software (GraphPad Software Corp, San Diego, CA, USA) was used for data analysis. 

### 2.10. Hemolytic Activity

The hemolytic activity of the Mf-Ag_2_ONPs was assessed following a previously established protocol [23]. Briefly, ten milliliters of defibrinated sheep blood was subjected to three consecutive washes with PBS 1x. Following these washes, a 1% erythrocyte suspension in PBS 1x was prepared. This erythrocyte suspension was subsequently mixed in a 1:1 ratio with Mf-Ag_2_ONPs, positive controls (10% Triton X-100), or negative controls (PBS 1x) in a 96-well polypropylene plate. The mixture was incubated at 37 °C for 1 h. Post-incubation, the samples were centrifuged for 5 min at 1700× *g*. The supernatant was then carefully transferred to a transparent flat bottom 96-well plate for absorbance measurement at 405 nm using a Cytation5 multi-mode plate reader (BioTek). Each experiment included three technical replicates, and the entire procedure was repeated three times. For each sample, the hemolysis rate was calculated according to the formula:$$HR\left(\%\right)=\frac{OD_{test}-OD_{neg}}{OD_{pos}-OD_{neg}}*100$$

### 2.11. Statistical Analysis

Data for all experiments were obtained in triplicates, and the mean ± standard deviation (SD) was obtained. Two-way ANOVA Dunnett tests determined the significance of mean differences across groups using the GraphPad Prism 10.2 software (GraphPad Software Corp, San Diego, CA, USA) for the Biofilm Inhibition Evaluation. The *p* values < 0.05, <0.01, and <0.001 were considered statistically significant.

## 3. Results

### 3.1. Quantification of Phenolic Compounds

The total phenolic content of the aqueous extract of *M. flexuosa* was determined using the Folin–Ciocalteu assay and expressed as GAE equivalents, resulting in 141.9 ± 6.8 mg GAE/100 g of dry extract. Additionally, high-performance liquid chromatography analysis of these aqueous extracts presented a total phenol concentration of 1609.2 mg/100 g dry weight as a sum of the individual compounds (Table 1). 

### 3.2. Phyto-Fabrication Mf-Ag_2_ONPs

The addition of an aqueous Mf extract to a colorless silver nitrate solution resulted in a color change from yellow to reddish-brown within 90 min, indicating the formation of Mf-Ag_2_ONPs. UV–visible spectroscopy further confirmed the formation of Mf-Ag_2_ONPs, showing an absorbance peak around 460 nm, indicative of silver oxide nanoparticle presence (Figure 1a,b). 

### 3.3. Characterization of Silver Oxide Nanoparticles 

#### 3.3.1. Dynamic Light Scattering (DLS) and Zeta Potential

The hydrodynamic diameter of the nanoparticles measured after the synthesis process was around 134.4 ± 39.5 nm (Figure 2). The calculated average of nine measurement values of ζ-potential for the nanoparticles was around −10.2 mV.

#### 3.3.2. X-Ray Diffraction (XRD)

The XRD diffractogram for Mf-Ag_2_ONPs is shown in Figure 3. The pattern is consistent with previously reported data on green-synthesized Ag_2_ONPs [29]. The identified peak is characteristic of a cubic silver oxide structure and corresponds to the most intense reflection plane (1 1 1) of silver oxide. A minor peak is observed at around 38°, which could be associated with a silver or a silver oxide structure. Nevertheless, the main product of synthesis is silver oxide nanoparticles. These data are in concordance with the standard data of Crystallography Open Database 1010486. Using the Scherrer equation, the crystallite grain size is estimated to be around 110 nm, in line with DLS results. 

#### 3.3.3. Transmission Electron Microscopy (TEM)

TEM observations were performed to identify the size and morphology of the Mf-Ag_2_ONPs (Figure 4). The Mf-Ag_2_ONPs showed a quasi-spherical shape with an average diameter of about 25.7 ± 8.7 nm. Compared with DLS and XRD studies, the result obtained via the TEM image evidences a smaller size of the Mf-Ag_2_ONPs. This is related to the presence of the extract around the Ag_2_ONPs, which can be easily identified in Figure 4. It also shows the polydisperse nature of the synthesized nanoparticles.

#### 3.3.4. Energy-Dispersive X-Ray Spectroscopy (EDS) and Scanning Electron Microscopy (SEM) 

EDS analysis confirms the presence of Ag (73.6 norm. wt.%), O (18.1 norm. wt.%), and other elements of the organic portion (Figure 5a). This elemental composition corroborates the results obtained by XRD, thereby confirming that the product obtained in the synthesis is Ag_2_O. The SEM image is unable to demonstrate the morphology of the Mf-Ag_2_ONPs, which is likely due to the high presence of *M. flexuosa* extract. Nevertheless, it can be observed that the nanoparticles exhibit a homogeneous distribution (Figure 5b). 

#### 3.3.5. FTIR

FTIR was used to analyze the structure of phytochemicals presented in aqueous leaf extracts of *M. flexuosa*, which are responsible for surface coating and Ag_2_ONP stabilization. Figure 6 shows the IR spectra of the aqueous extract and the synthesized Mf-Ag_2_ONPs.

### 3.4. Antibacterial Activity

The determination of MIC is fundamental in monitoring resistance development and establishing optimal pharmaco-dynamic dosing. The antibacterial activity of Mf-Ag_2_ONPs was evaluated against four non-MDR-resistant bacteria and five multidrug-resistant strains from clinical isolates. The MIC values for each bacterial strain are summarized in Table 2. The highest MIC value of 45 µg/mL was recorded for *E. faecium*, indicating lower sensitivity to the Mf-Ag_2_ONPs than the other bacterial strains. MIC values of 22.5 µg/mL were observed for Gram-positive bacteria *S. aureus* ATCC 25923, Gram-negative bacteria *E. coli* ATCC 25922, and MDR strains *K. pneumoniae*, *E. coli*, *S. enterica* serovar Kentucky, and *P. aeruginosa*. The Gram-negative bacteria *P. aeruginosa* ATCC 27853 and *B. cepacia* ATCC 25416 exhibited the lowest MIC values at 11.25 µg/mL, indicating higher sensitivity to the Mf-Ag_2_ONPs.

### 3.5. Antifungal Activity

The antifungal activity of Mf-Ag_2_ONPs was assessed against various *Candida* strains. The MIC values, measured in micrograms per milliliter (µg/mL), quantitatively evaluate the nanoparticles’ efficacy in inhibiting fungal growth.

For *C. krusei* ATCC 14243, the MIC was determined to be 11.25 µg/mL, indicating the concentration of Mf-Ag_2_ONPs required to inhibit the growth of this strain. Similarly, *C. albicans* ATCC 10231 exhibited an MIC of 11.25 µg/mL, demonstrating comparable sensitivity to Mf-Ag_2_ONPs as *C. krusei*. In contrast, *C. glabrata* ATCC 66032 showed a significantly lower MIC value of 5.63 µg/mL, suggesting higher susceptibility to the antifungal effects of the nanoparticles and highlighting the potential effectiveness of *M. flexuosa*-coated Ag_2_ONPs against infections caused by this strain. However, *C. tropicalis* ATCC 13803 did not yield a definitive MIC value (ND), indicating that Mf-Ag_2_ONPs were ineffective in inhibiting this strain’s growth at the tested concentrations.

Further investigation into the MIC values across different strains revealed notable variations in susceptibility. The consistent MIC of 11.25 µg/mL for both *Candida krusei* and *C. albicans* suggests a similar interaction mechanism with the nanoparticles, which could be related to the structural similarities in their cell walls.

The inability to determine a definitive MIC for *C. tropicalis* implies that the concentrations tested were insufficient to inhibit this strain or that the strain possesses inherent resistance mechanisms against the nanoparticles. This highlights the necessity for additional studies to explore higher concentrations or alternative formulations to achieve antifungal efficacy against *C. tropicalis*.

The results of the MIC assay are summarized in Table 3, providing a clear comparison of the antifungal activity of Mf-Ag_2_ONPs across the different *Candida* strains. The data indicate a concentration-dependent inhibition of fungal growth, with varying degrees of susceptibility observed among the strains tested.

These findings underscore the potential of Mf-Ag_2_ONPs as effective antifungal agents, particularly against *C. glabrata*. The notable efficacy at lower MIC values suggests that these nanoparticles could be further developed for clinical applications, offering an alternative to conventional antifungal treatments. The differential responses observed among the Candida strains warrant further investigation to optimize the use of Mf-Ag_2_ONPs and explore their full therapeutic potential.

### 3.6. Biofilm Inhibition Activity

This study tested the biofilm inhibition effects of Mf-Ag_2_ONPs (2.5–40 µg/mL) by crystal violet staining. The MBIC (minimum biofilm inhibitory concentration) to inhibit at least 50% biofilm growth was considered for the statistical analysis. The biofilm formation of three bacterial strains (*S. aureus* ATCC 25923, *P. aeruginosa* ATCC 9027, and *B. cepacia* ATCC 25416) was statistically significantly inhibited by nanoparticle treatment at 40 and 20 µg/mL Mf-Ag_2_ONP concentrations. The biofilm formation of the bacterial strain *P. aeruginosa* ATCC 9027 was the most significantly inhibited after treatment with 20 µg/mL Mf-Ag_2_ONPs with an inhibition rate of 75% ± 16 (*p* = 0.001). The biofilm formation of *B. cepacia* ATCC 25416 showed the highest percentage of inhibition at 20 µg/mL out of the three bacterial strains (87% ± 12). Additionally, the biofilm formation of *S. aureus* ATCC 25923 displayed an inhibition of 71% ± 5 after treatment with 20 µg/mL Mf-Ag_2_ONPs (Figure 7). The inhibition percentage of *L. monocytogenes* ATCC 13932 was lower than 50% after treatment with the highest concentration of Mf-Ag_2_ONPs tested during this study (40 µg/mL); as a result, it was considered non-active for our MBIC_50_ report. The MBIC_50_ and inhibition rate percentages can be found in Appendix A. 

### 3.7. Antioxidant Activity

The antioxidant activity of the nanoparticles (Mf-Ag_2_ONPs) was evaluated by the DPPH assay and the ABTS/TEAC assays, as shown in Table 4. The nanoparticles resuspended in water had a 9-fold difference in the IC_50_ values between the DPPH and the ABTS assays; the IC_50_ values for the Mf extract varied by up to 16-fold, and the IC_50_ values for the control had an 11-fold difference. In all the cases, the IC_50_ was lower for the DPPH assay. Additionally, TEAC values were estimated to allow a relative quantification of the antioxidant properties of the Mf-Ag_2_ONPs based on the Trolox standard. 

The ascorbic acid standard’s radical scavenging activity was 2- to 3-fold more active than the Mf-Ag_2_ONPs, based on the DPPH and the ABTS/TEAC assays. On the other hand, the Mf extract showed lower antioxidant properties than the Mf-Ag_2_ONPs and had at least a 15-fold higher IC_50_. Together, these results underscore the exceptional antioxidant capabilities of Mf-Ag_2_ONPs. 

### 3.8. Anticancer Activity

The findings illustrate the anti-proliferative effects of Mf-Ag_2_ONPs on a range of tumor cell lines (HeLa, HCT116, THJ29T, and MDA-MB-231) and a non-tumor cell line (NIH3T3). Cellular proliferation assessed following a 72 h exposure to varying concentrations of Mf-Ag_2_ONPs and Mf extract demonstrated dose-dependent inhibition (Figure 8 and Appendix A). 

The calculated inhibitory concentration values (IC_50_) using dose–response curves (Table 5) revealed significant anti-proliferative activity. Mf-Ag_2_ONPs exhibited strong potency across all tumor cells, with IC_50_ values ranging from 3.5 to 10.8 μg/mL. This indicates a robust inhibition of cell proliferation, especially in HCT116 and HeLa cells, which were more sensitive to the nanoparticles. In contrast, Mf extract displayed IC_50_ values between 1336 and 4048 μg/mL, indicating reduced effectiveness. The comparison of IC_50_ values showed a significant difference between Mf-Ag_2_ONPs and Mf extract, with the nanoparticles being 300–600 times more potent in tumor cells than the extract. When compared to CDDP used as a positive control, the IC_50_ values of Mf-Ag_2_ONPs demonstrated similar or slightly higher IC_50_ values, between 2.3 and 10.6 μg/mL. Despite this, the therapeutic index (TI) for Mf-Ag_2_ONPs suggests a comparable or even more favorable safety profile compared to CDDP, particularly in HCT116 and THJ29T.

Among the tumor cell lines studied, colorectal cancer (HCT116) was the most sensitive, while breast cancer (MDA-MB-231) was the most resistant. These results indicate that Mf-Ag_2_ONPs have a substantial dose-dependent and anti-proliferative effect across multiple cell lines. 

### 3.9. Hemolytic Activity

Hemolytic activity of Mf-Ag_2_ONPs, measured as a percentage, was assessed across various nanoparticle concentrations and compared against established positive and negative controls, as seen in Table 6. The negative control (PBS 1x, labeled C−) exhibited 0% hemolytic activity, indicating no disruption of red blood cell (RBC) membranes under the test conditions. This serves as a critical reference point, confirming that the test conditions themselves do not induce hemolysis. Conversely, the positive control (10% Triton X-100, labeled C+) exhibited 100% hemolytic activity, validating the assay’s capability to detect hemolysis.

For the test samples, hemolytic activity demonstrated a concentration-dependent increase. At the lowest concentration tested, 10 µg/mL, hemolytic activity was minimal at 0.2% (Table 6). This negligible hemolysis suggests that Mf-Ag_2_ONPs are highly biocompatible with erythrocytes at lower concentrations, posing minimal risk of cytotoxicity in this range. The slight hemolytic activity observed indicates that Mf-Ag_2_ONPs do not significantly disrupt RBC membranes at this concentration. As the concentration increased to 20 µg/mL, hemolytic activity rose to 1.7% (Table 6). This gradual increase in hemolytic potential implies the beginning of a dose-dependent response, wherein higher concentrations of Mf-Ag_2_ONPs start to interact more noticeably with RBC membranes. However, the hemolytic activity at this concentration remains relatively low, suggesting that the nanoparticles are still largely biocompatible and that their interaction with RBCs is not severely cytotoxic.

At 40 µg/mL, hemolytic activity further increased to 3.9% (Table 6). This level of RBC lysis indicates a moderate degree of membrane disruption. Despite this increase, the hemolytic activity at this highest concentration tested is still substantially lower than the positive control, which exhibited complete hemolysis. This comparison underscores the fact that even at elevated concentrations, Mf-Ag_2_ONPs do not induce extensive hemolysis. The moderate hemolytic activity observed suggests a threshold beyond which the nanoparticles begin to exert more significant cytotoxic effects, but still within a range that is considerably less harmful than known cytotoxic agents such as Triton X-100.

These observations highlight the concentration-dependent nature of Mf-Ag_2_ONP-induced hemolysis, with a clear trend of increasing hemolytic activity at higher nanoparticle concentrations. The critical finding here is that Mf-Ag_2_ONPs, even at the highest concentration tested, remain below the hemolytic threshold of the positive control, indicating a relatively safe profile for biomedical applications where minimal hemolysis is crucial.

## 4. Discussion

### 4.1. Identification of Bioactive Compounds in the Extract and Phyto-Fabrication Mf-Ag_2_ONPs

In the current study, “aguaje” fruit extract was employed to synthesize Ag_2_ONPs, acting as a bio-reductant and stabilizing agent. The phenolic profile of the aqueous solution of *M. flexuosa* showed compounds such as *m*-coumaric acid, procatechin acid, syringic acid, gallic acid, naringenin, and protocatechuic acid [30,31,32]. These compounds and other biomolecules, acting as essential reducing or capping agents, facilitate the synthesis of nanoparticles [33].

Upon combining the plant extract with the precursor salt, the color of the solution undergoes a discernible change from yellow to reddish brown after incubation. This color transformation is due to surface plasmon resonance (SPR), which confirms the reduction process facilitated by the plant’s metabolites. SPR occurs when electromagnetic radiation interacts with the nanoparticle’s surface, resulting in the observed color shift [34].

### 4.2. Characterization of the Phytosynthesis of the Mf-Ag_2_ONPs

The synthesis of silver oxide nanoparticles was confirmed at 460 nm in the UV-Vis spectrum. The findings of our research utilizing UV-Vis spectroscopy are in accordance with those of an earlier study that used a variety of plant extracts [35,36].

The ζ-potential of the nanoparticles was around −10.2 mV., meaning their surfaces possess negative charges under the experimental conditions (i.e., aqueous media, pH 5.1), indicating that a moderate electrostatic repulsion takes place among them. These negative charges could come from the hydroxyl groups of the phenolic compounds. As reported in the literature, nanoparticles having the measured absolute ζ-potential indicates that stability could be in the range of incipient instability [37]; i.e., Van der Waals forces could dominate the solution and generate flocculation/aggregation, which was not observed in the nanoparticle solution within a 15-day period.

XRD revealed that the synthesis is mainly silver oxide, with a few quantities (less than 1%) of Ag^0^ nanoparticles. XRD and DLS gave nearly the same crystallite size, around 110 nm, while 25 nm is calculated from TEM measurements. One can conclude that the extract coating size can be approximated to 85 nm.

In the FTIR spectrum (Figure 6), an intense and broad signal can be observed between 3600 cm^−1^ and 3200 cm^−1^, attributed to the stretching vibrations of the O–H group. In the 2800–3000 cm^−1^ range, the signals observed correspond to asymmetric and symmetric stretching vibrations of aliphatic C–H [38]. The bands observed in the region 1700–1500 cm^−1^ are assigned to the C=O and C=C stretching vibrations. The peak at 1692 cm^−1^ is attributed to the vibration of unsaturated carboxylic acids, while the band at 1647 cm^−1^ is associated with stretching the double bond C=C present.

Signals around 1400 cm^−1^ are related to CH_2_ scissor deformation vibration, while those between 1200 cm^−1^ and 800 cm^−1^ are attributed to the polysaccharide fingerprint [39]. Furthermore, the band at 1072 cm^−1^ suggests the presence of sugars such as arabinose, galactose, and xylose, while the signals at 1148 cm^−1^ and 1049 cm^−1^ are attributable to the axial deformation of C–O and may be associated with the presence of galacturonan derivatives, as previously reported [39].

A slight shoulder is seen at 1726 cm^−1^ when the carbonyl region is closely examined. This shoulder, along with the signals at 1356 cm^−1^ and 1236 cm^−1^, can be attributed to the axial deformation of the ester groups C=O, -C-CH_3_, and -C-O-, respectively [39], and possibly to the ester groups present in the pectic polysaccharides that have been reported in the aqueous extracts of *M. flexuosa* [40].

### 4.3. Antibacterial Activity

Antimicrobial susceptibility testing is essential for the efficacious management of pathogenic microorganisms. In the current study, Mf-Ag_2_ONPs exhibited potent inhibitory effects against the non-MDR strains *P. aeruginosa* and *B. cepacia* with MIC values as low as 11.25 µg/mL, suggesting that these Gram-negative bacteria are particularly susceptible to Mf-Ag_2_ONPs. Studies suggest higher efficacy of nanoparticles against Gram-negative bacteria due to the thickness and cell wall composition [41,42]. This may indicate that Gram-negative bacteria are generally more susceptible, whereas Gram-positive bacteria may demonstrate a certain level of resistance [43]. However, our results show higher MIC values for other Gram-negative bacteria, indicating that the antibacterial effectiveness of Ag_2_ONPs synthesized from *M. flexuosa* varies across Gram-negative strains. It is worth noting that Mf-Ag_2_ONPs were effective against both Gram-negative and Gram-positive strains.

The MIC values for the other non-MDR strains, *E. coli* ATCC 25922 and *S. aureus* ATCC 25923, also indicate that Mf-Ag_2_ONPs possess significant antimicrobial activity with an MIC value of 22.5 µg/mL. Abbasi showed the antibacterial efficacy of silver oxide nanoparticles synthesized with *Rhamnus virgata* leaf extract against *E.coli* and *P. aeruginosa*, reporting MIC values ranging from 28.12 to 225 µg/mL, which are comparable to our study [44]. Elemike (2017) reported an MIC for *S. aureus* of 75 µg/mL using Ag/Ag_2_O nanoparticles synthesized from the aqueous leaf extract of *Eupatorium odoratum* [43].

In the case of MDR strains such as *K. pneumoniae*, *E. coli*, *S. enterica serovar Kentucky*, and *P. aeruginosa*, the Mf-Ag_2_ONPs showed significant antimicrobial activity. The MICs for these strains were 22.5 µg/mL. In contrast, *E. faecium*, a Gram-positive strain, displayed the highest MIC at 45.0 µg/mL. This suggests that while Mf-Ag_2_ONPs are effective, their efficacy may be reduced against certain MDR bacteria perhaps due to their Gram-positive nature.

There is little information on the use of Ag_2_ONPs obtained from aqueous extracts against multi-resistant bacteria. Therefore, this study provides new insights on the use of silver oxide nanoparticles in these multi-resistant bacterial strains.

The antibacterial efficiency of silver oxide nanoparticles could be related to the features of the bacterial species and physicochemical properties of nanoparticles, including size and surface or other resistance mechanisms that decrease the nanoparticles’ access to intracellular targets [45]. Possible antibacterial mechanisms have been suggested by multiple studies. Ag_2_ONPs are believed to interact strongly with sulfur- and phosphorus-containing molecules, such as DNA, leading to impaired DNA replication [46]. Another potential mechanism for silver oxide nanoparticles binding to the bacterial membrane involves electrostatic attraction and interaction with sulfur-containing proteins located on the bacterial membrane. This interaction may compromise the integrity of the cell envelope increasing the permeability of the cell membrane [44]. Another possible bactericidal effect of silver oxide nanoparticles is the release of silver ions after penetration. The release of atomic Ag^0^ and ionic Ag^+^ clusters leads to cell death by inhibiting a respiratory enzyme [33,47]. Some studies indicate that the antibacterial activity of nanoparticles could be due to the adsorbed functional groups on the surface of the nanoparticles [48]. Earlier studies have discussed the bactericidal potential of silver oxide nanoparticles and shown that ROS generation is the core mechanism that provides antimicrobial potency to the nanoparticles. ROS generation triggers cell oxidative stress and the induction of toxicity [49].

Overall, our study underscores the potential of Ag_2_ONPs synthesized from *M. flexuosa* fruit extract and provides evidence to support the use of these nanoparticles as an alternative antimicrobial strategy against Gram-positive and Gram-negative bacteria, including MDR strains.

### 4.4. Antifungal Activity

*Candida* spp. is the most common opportunistic fungal pathogen in humans, frequently occurring as a hospital-acquired infection and ranking as the fourth leading cause of nosocomial infection [50]. Non-*albicans* species, including *C. tropicalis*, *C. glabrata*, and *C. parapsilosis*, significantly contribute to candidiasis. *C. glabrata* is particularly notable, being the second or third most common cause of this infection after *C. albicans*. The growing resistance of fungi to existing antibiotics is a major concern, underscoring the importance of developing novel antifungals as alternative treatments to conventional antifungals and addressing the escalating issue of drug resistance [51].

Our findings align with previous research demonstrating the antifungal activity of silver nanoparticles synthesized with plant extracts against *Candida* species, supporting the potential of plant extract-mediated nanoparticles as effective antifungal agents. For instance, Shahzad and coworkers evaluated silver oxide nanoparticles using *Mentha pulegium* and *Ficus carica* extracts against *C. albicans* using a disk diffusion technique using different concentrations of the samples (12.50, 25, and 50 mg/mL). The results indicated that the nanoparticles exhibited significant antifungal activity against Candida albicans, as evidenced by the 10mm inhibition zones [52]

Recently, the MIC of ovalbumin-mediated silver oxide nanoparticles was assessed, resulting in moderate antifungal activity against *C. albicans* and *Candida parapsilosis* with an MIC of 25 µg/mL [53]. In contrast, our study demonstrates a significantly lower MIC of 11.25 µg/mL, which is half the dose reported by Kumar et al., indicating a much better antifungal efficacy that could be more favorable for potential therapeutic applications. Researchers also assessed the antifungal activity of silver nanoparticles (AgNPs) against clinical isolates of *Candida auris.* The MIC of AgNPs was less than 6.25 μg/mL [54], which is comparable to our findings that report MIC values ranging from 5 to 11 μg/mL for different *Candida* species.

In our research, silver nanoparticles demonstrate significantly higher antifungal activity against *C. krusei*, *C. albicans*, and *C. glabrata* compared to *C. tropicalis*. The lowest MIC value observed was 5.625 µg/mL against *C. glabrata*, suggesting that the silver nanoparticles synthesized with *M. flexuosa* fruit extract possess substantial antifungal potential, comparable to nanoparticles synthesized with other plant extracts.

The antifungal mechanisms of nanoparticles are multifaceted and involve several biochemical and cellular processes. Lipovsky et al. observed that smaller ZnO NPs exhibited better antifungal activity against *C. albicans* than larger particles [55]. The anticandidal activity of ZnO NPs is attributed to the intracellular production of free radicals such as hydroxyl radicals, singlet oxygen radicals, superoxide radicals, and nitric oxide radicals, which can cross the nuclear membrane and cause DNA damage, leading to irreversible chromosome damage or cell death [56,57]. The fungicidal activity is proposed to result from the formation of insoluble compounds by the inactivation of sulfhydryl groups in the fungal cell wall, disrupting membrane-bound enzymes and lipids, ultimately causing cell death [58]. These results suggest that ZnO NPs may provide a novel family of fungicidal compounds to treat candidiasis.

The antifungal activity of our synthesized silver nanoparticles, particularly against *C. glabrata*, is promising and aligns with findings from other studies using plant extracts for nanoparticle synthesis. These results suggest potential clinical applications in treating fungal infections, especially in cases where conventional treatments are less effective due to resistance.

### 4.5. Biofilm Inhibition Activity

Biofilms pose significant health challenges, especially when formed on medical devices like urinary catheters, leading to persistent infections. They are notoriously resistant to antimicrobial agents compared to planktonic cells, prompting extensive research into effective biofilm control strategies. Studies have consistently shown that both chemically synthesized and green-synthesized Ag_2_ONPs can play a relevant role as biofilm inhibitors [44].

In our research, the biofilm inhibitory effects of Mf-Ag_2_ONPs (ranging from 2.5 to 40 µg/mL) and the *M. flexuosa* fruit extract were evaluated using crystal violet staining to determine the MBIC. Significant biofilm inhibition was observed at the two highest concentrations of Mf-Ag_2_ONPs (40 µg/mL and 20 µg/mL), against *S. aureus*, *B. cepacia,* and *P. aeruginosa*. The literature regarding the assessment of silver oxide nanoparticles against bacterial biofilms is scarce. In 2021, a study described the positive biofilm inhibition effects of bio-fabricated Ag_2_ONPs in a Gram-positive *Bacillus subtilis* strain, which was grown over a glass slide [59].

Other studies have also assessed the biofilm inhibitory effects of Ag_2_ONPs in Gram-negative species of high clinical significance like *E. coli* and *Klebsiella Pneumonia* [60,61]. In our study, both of our biofilm Gram-negative strains, namely *P. aeruginosa* and *B. cepacia*, displayed the highest inhibition rates after treatment with 20 µg/mL of Ag_2_ONPs. Currently, there are no recorded studies of the use of green-synthesized Ag_2_ONPs against these specific strains. The study by Rajivgandhi et al. (2021) showed that around 100 µg/mL of floral green-synthesized Ag_2_ONPs was required for a 50% biofilm inhibition rate of their Gram-negative strain, *K. pneumoniae* [60]. This concentration is importantly higher than the one recorded in our study. In this regard, this small comparison can serve as a starting point to encourage the green synthesis of Ag_2_ONPs with different plant extracts that could end up yielding better biofilm inhibitory results.

In the same line, there is a lack of research regarding the potential mechanisms of biofilm inhibition of Ag_2_ONPs. Some suggestions can be taken from the mechanisms described for similar silver nanoparticles like AgNPs. Some of the observed mechanisms for AgNPs include the inhibition of biofilm formation by disrupting the initial adherence of cells to surfaces, a critical step in the development of biofilms. Some studies have demonstrated that AgNPs neutralize adhesive substances essential for biofilm formation and can also mediate bacterial apoptosis by disrupting the bacterial actin cytoskeleton [62,63]. Additionally, the accumulation of AgNPs and liberation of silver ions compromise bilayer integrity and stop the interaction of bacterial cells before they accumulate and form biofilm structures [64]. Moreover, AgNPs display potent antibacterial activity in planktonic bacterial cultures by interfering with bacterial biological processes [65]. In this regard, we can extrapolate that bacterial toxicity displayed by silver-derived nanoparticles like AgNPs or our Ag_2_ONPs could inhibit the formation of more complex community structures, namely biofilms, because of the systemic damage and stress applied over individual cells.

Overall, our study underscores the significant antibiofilm potential of Mf-Ag_2_ONPs synthesized from *M. flexuosa* fruit extract and provides evidence to support using this extract as an alternative antimicrobial strategy. Additionally, it strongly encourages the study of green-synthesized Ag_2_ONPs and their mechanisms of antibiofilm activity as they showcase strong promise as antibiofilm tools.

### 4.6. Antioxidant Activity

Antioxidant nanoparticles reduce oxidative stress, enhancing disease treatment, drug stability, and anti-aging effects. They improve biocompatibility, minimize inflammation, and help preserve food quality by preventing degradation [66,67]. Therefore, finding nanoparticles with antioxidant properties is important.

As reported in the results, the IC_50_ was lower for the DPPH assay than the ABTS/TEAC assays. A similar trend was observed in the study by Hu et al. (2022) where DPPH IC_50_ values (11.75 µg/mL) of the silver nanoparticles synthesized using the *Areca catechu* nut aqueous extract, a member of the *Arecaceae* family, were lower than the ABTS IC_50_ values (44.85 µg/mL) [68]. This could be attributed to the nature of the assay or the fact that some antioxidants react quickly and completely, while others react slowly [69]. Notably, the reaction times for the ABTS/TEAC and the DPPH assays differ in accordance with their established protocols [26,27]. In the case of the ABTS, it is possible that the 5 min of incubation was insufficient to achieve the maximum decolorization of the ABTS radical solution. On the other hand, 40 min of incubation for the DPPH assay could have been enough to scavenge all the DPPH radicals. 

According to the DPPH and ABTS/TEAC assays, the ascorbic acid standard’s radical scavenging activity was at least 2-fold more active than the Mf-Ag_2_ONPs. Yet, the Mf-Ag_2_ONPs’ DPPH IC_50_ was 14.77 µg/mL, suggesting high antioxidant potential [68]. A study by Hu et al. (2022) reported a similar IC_50_ of 11.75 µg/mL for AgNPs obtained using an aqueous extract of another plant from the Arecaceae family [68]. However, Das et al. reported a higher IC_50_ value (96.39 µg/mL) for AgNPs obtained from *Cocos nucifera* extract [70]. In the case of Ag_2_ONPs reported from different plant families such as *Callistemon lanceolatus* (aqueous leaf extract), an IC_50_ of 62.12 µg/mL was attained [71], whereas an IC_50_ of 618.21 μg/mL for *Nigella sativa* seeds (aqueous extract) was calculated [72]. Other examples include the production of Ag_2_ONPs using *Thunbergia mysorensis* stems and flowers (aqueous extract) with an IC_50_ of 17.02 μg/mL for the flowers and an IC_50_ of 32.15 μg/mL for the stem. The latter study also found an ABTS IC_50_ of 148.02 μg/mL for the flowers and an IC_50_ of 178 μg/mL for the stem-produced Ag_2_ONPs [73], which are in a similar range with our findings’ IC_50_ of 138.50 μg/mL. Overall, these results highlight the antioxidant capabilities of the Mf-Ag_2_ONPs (this study). It is worth noting that there are no reports of the antioxidant activity of Ag_2_ONPs obtained using the *M. flexuosa* extract.

On the other hand, in our study, the fruit extract (Mf extract) showed lower antioxidant properties and had a 15- to 25-fold higher IC_50_ (227.40 µg/mL DPPH and 3596.67 µg/mL ABTS) than the Mf-Ag_2_ONPs. Even though the Mf extract’s IC_50_ values were high compared to the Mf-Ag_2_ONPs’, another study by Koolen et al. (2013) reported an even higher DPPH IC_50_ value of 19.58 mg/mL for the extract obtained from green fruits of *M. flexuosa* [74]. Importantly, in our study, mature fruits of *M. flexuosa* were used to obtain the extract, which could contribute to this difference. Research has indicated that *M. flexuosa* pulp is rich in antioxidants, including carotenoids, tocopherols, ascorbic acid, and phenolic compounds [19]. Yet, the composition and levels can vary depending on the green (unripe) and mature (ripe) stages of the fruit, the morphotype of the fruit, as well as the growth location [19,75,76]. Additionally, the extraction method influences the yield of bioactive molecules. In our study, this process was carried out using only water in contrast to other reports, which used various solvents [19,75,76], and the total polyphenolic content was 16.6 ± 0.8 µg GAE/mL of extract.

Notably, the TEAC values revealed a similar trend between the Mf-Ag_2_ONPs (3468.67 µmol TE/g) and the Mf extract (123.16 µmol TE/g), achieving a 28-fold difference in antioxidant capacity. Similar studies have found TEAC values ranging from 33.02 to 70.2 µmol TE/g for the fruit extract [75,77], suggesting that our Mf extract had a higher antioxidant capacity. However, information is not available for Ag_2_ONPs’ TEAC. Overall, the intrinsic antioxidant potential of the fruit extract, as seen in different studies, suggests an important contribution to the resulting Mf-Ag_2_ONPs’ antioxidant potential.

Together, these results underscore the exceptional antioxidant capabilities of Mf-Ag_2_ONPs. The antioxidant capabilities of silver oxide nanoparticles made from *M. flexuosa* are due to the fruit extract’s abundance of bioactive elements, like carotenoids, tocopherols, ascorbic acid, and phenolic compounds, which are well known for their strong antioxidant properties [78]. These elements work together to amplify the Mf-Ag_2_ONPs’ antioxidant potential during synthesis. Nanoparticles with these features are essential in fighting diseases caused by oxidative stress and shielding cells from damage caused by free radicals [79].

### 4.7. Anticancer Activity

The efficacy of Mf-Ag_2_ONPs and Mf extract against cancer cells reflects a range of factors influencing their effectiveness and safety. The improved efficacy of Mf-Ag_2_ONPs can be attributed to their increased bioavailability and cellular uptake. This aligns with various studies suggesting that nanoparticles enhance drug delivery and therapeutic effects by accumulating more effectively in tumor tissues and reducing off-target side effects [80,81]. Various studies have demonstrated the significant anticancer properties of biosynthesized Ag_2_ONPs across different cancer cell lines [33]. Notably, both AgNPs and AgONPs synthesized from plant sources have exhibited similar anticancer profiles, primarily through mechanisms involving the activation of reactive oxygen species, cytological aberrations, oxidative stress, apoptosis, and subsequent cell membrane degradation [82,83,84,85].

However, it is important to consider that the presence of biomolecules from natural extracts used in the synthesis process can increase the size of these nanoparticles [86], potentially making cell membrane penetration more challenging and affecting their overall effectiveness [84,87]. Despite this, our findings indicate that even with a 5-fold increase in size due to the stabilizing Mf extract corona, the IC_50_ values for Mf-Ag_2_ONPs remained low, comparable to, or even better than other Ag_2_ONPs [88,89,90]. This suggests that the efficacy of these nanoparticles remains high, regardless of their increased size.

Additionally, several studies show that green-synthesized AgNPs and other metal nanoparticles are less genotoxic than chemically synthesized ones [91,92] and are more cytotoxic to cancer cells due to cancer cells’ defective defenses and resistance to apoptosis [93,94,95,96].

While Mf-Ag_2_ONPs demonstrated superior anticancer efficacy compared to Mf extract, the TI values were higher for Mf extract, indicating potentially fewer side effects. The higher TI for both Mf-Ag_2_ONPs and Mf extract on HeLa and HCT116 cells suggests their potential use in cervical and colorectal cancers. Other AgONPs displayed better safety profiles by selectively targeting cancer cells while sparing normal cells [83], in some cases showing a 4-fold difference in activity between cancerous and normal cells [35]. The physical attributes and surface properties of green-synthesized Mf-Ag_2_ONPs, along with the biomolecular interactions during synthesis, may enhance their uptake into non-tumor cells, potentially explaining their observed toxicity [97]. Further research is needed to fully understand these interactions and optimize the therapeutic potential of these nanoparticles. 

In conclusion, the unique physicochemical properties of Mf-Ag_2_ONPs, combined with the bioactive compounds present in natural extracts, contribute to the anticancer effects [98,99]. The anticancer activity of *M. flexuosa* fruit extract can be attributed to the combined effects of its flavonoids and phenolic acids, such as the ones identified in this article (gallic acid, protocatechuic acid, syringic acid, m-coumaric acid, naringenin, and p-coumaric acid), which are known for their anticancer properties [100,101]. The synergy between these bioactive compounds collectively contributes to the extract’s overall effectiveness inhibiting cancer cell proliferation and inducing apoptosis. Additionally, fatty acids and carotenoids are among the primary nutrients found in *M. flexuosa* [102,103]. Specifically, its polyunsaturated fatty acids have the potential to inhibit the growth of cancer cells and initiate apoptosis. In addition, carotenoids act as powerful antioxidants, defending cells from damage caused by free radicals, modulating the immune system, and hindering the growth of cancer cells [19,104]. 

In summary, while Mf-Ag_2_ONPs display substantial dose-dependent anti-proliferative effects across multiple cell lines, the observed differential response between tumor and non-tumor cells necessitates further investigation into the underlying mechanisms and potential therapeutic implications. 

### 4.8. Hemolytic Activity

Our observations of hemolytic activity align with the existing literature, emphasizing the importance of nanoparticle concentrations in determining cytotoxic effects. Minimal hemolytic activity at 10 mg/mL (0.2%) is consistent with findings that biogenic Ag_2_ONPs synthesized using plant extracts generally exhibit lower toxicity compared to chemically synthesized counterparts. For instance, Ag_2_ONPs synthesized with *Carica papaya* and *Trigonella foenum-graecum* aqueous extracts demonstrated lower hemolytic activity than their chemically synthesized counterparts with respect to concentration [105].

The biocompatibility observed at lower concentrations of Mf-Ag_2_ONPs can be attributed to bioactive compounds in the extract, which may provide a protective effect against cytotoxicity. Previous studies have demonstrated that plant extracts used in biogenic nanoparticle synthesis can confer enhanced stability and reduced toxicity. For example, AgONPs synthesized with *Thunbergia mysorensis* stem and flower extracts showed reduced hemolytic activity [73]. Our hemolytic activity data for Mf-Ag_2_ONPs align with these observations, suggesting that natural antioxidants and bioactive compounds in the extract play a role in mitigating cytotoxic effects [60]. The moderate hemolytic activity observed at higher concentrations in our study aligns with findings from other studies on biogenic AgNPs. For instance, Ag_2_ONPs synthesized using *Parieteria alsinaefolia* leaf extract exhibited concentration-dependent hemolytic activity, with higher concentrations leading to increased RBC lysis [34]. This similarity underscores a common pattern of cytotoxic response to increasing nanoparticle concentrations, irrespective of the plant extract used. The significant gap between the hemolytic activity of our highest concentration (3.9% at 40 mg/mL) and the positive control (100%) indicates that while Mf-Ag_2_ONPs are not entirely inert, they do not cause extensive hemolysis. This finding is crucial for biomedical applications, suggesting that these nanoparticles, when used at appropriate concentrations, can minimize adverse effects on RBCs.

However, while our study provides valuable insights into the hemolytic activity of Mf-Ag_2_ONPs, further research is needed to fully understand the mechanisms underlying their biocompatibility and cytotoxicity. Detailed investigations into the interaction between the nanoparticles and cellular membranes, as well as the role of specific bioactive compounds in the extract, would provide a more comprehensive understanding. Additionally, our study focused on in vitro hemolytic activity, which may not fully capture the complexities of in vivo interactions. In vivo studies are necessary to evaluate the biocompatibility and safety of these nanoparticles in a physiological context. Previous research has shown that in vivo environments can significantly influence the behavior and toxicity of nanoparticles [106,107].

Moreover, while our findings suggest a concentration-dependent increase in hemolytic activity, the specific thresholds for safe and effective concentrations in various biomedical applications need to be established. This requires a multidisciplinary approach, integrating insights from nanotoxicology, pharmacology, and materials science to optimize the design and application of biogenic Ag_2_ONPs.

## 5. Conclusions

In this study, silver nanoparticles (Mf-Ag_2_ONPs) were successfully synthesized using the aqueous extract of *M. flexuosa* fruits, demonstrating their potential as an eco-friendly and sustainable alternative to conventional antimicrobial agents. The Mf-Ag_2_ONPs exhibited significant antibacterial and antifungal activities, effectively inhibiting both non-resistant and MDR bacterial strains and Candida species. Additionally, these nanoparticles showed potent biofilm inhibitory properties, antioxidant capabilities, and pronounced anticancer effects against various cancer cell lines, highlighting their broad-spectrum biological activities.

The characterization of Mf-Ag_2_ONPs confirmed their formation and stability, supported by UV–visible spectroscopy and FTIR analysis. The antimicrobial assays revealed varied MICs, indicating the enhanced efficacy of Mf-Ag_2_ONPs compared to the aqueous extract alone. The antifungal tests demonstrated selective efficacy, particularly against *C. glabrata*, with the lowest MIC value observed. Our biofilm inhibition studies further highlighted the potential of Mf-Ag_2_ONPs as biofilm inhibitors by demonstrating important biofilm inhibition percentages in several bacterial strains of clinical relevance.

Antioxidant assays using DPPH and ABTS/TEAC methods revealed the exceptional radical scavenging abilities of Mf-Ag_2_ONPs, attributed to the bioactive compounds in *M. flexuosa*. The anticancer assays indicated a strong dose-dependent inhibition of cell proliferation in various cancer cell lines, with Mf-Ag_2_ONPs showing significantly higher potency compared to the Mf extract, and the reduced IC_50_ values for Mf-Ag_2_ONPs compared to the Mf extract underscore the potential of these nanoparticles as a more potent anticancer agent, justifying further investigation into additional studies to understand their mode of action and therapeutic applications in cancer treatment. Further research is needed to refine their synthesis and reduce their toxicity to non-tumor cells. Regarding the hemolytic activity, assays demonstrated minimal cytotoxicity at therapeutic concentrations, confirming the biocompatibility of Mf-Ag_2_ONPs.

Overall, this study’s findings suggest that silver nanoparticles synthesized from *M. flexuosa* fruit extract possess substantial antimicrobial, antifungal, antioxidant, and anticancer properties, with potential applications in the biomedical and environmental fields. Further in vivo studies are warranted to fully harness their therapeutic potential and explore their mechanisms of action in greater detail.

## Figures and Tables

**Figure 1 nanomaterials-14-01875-f001:**
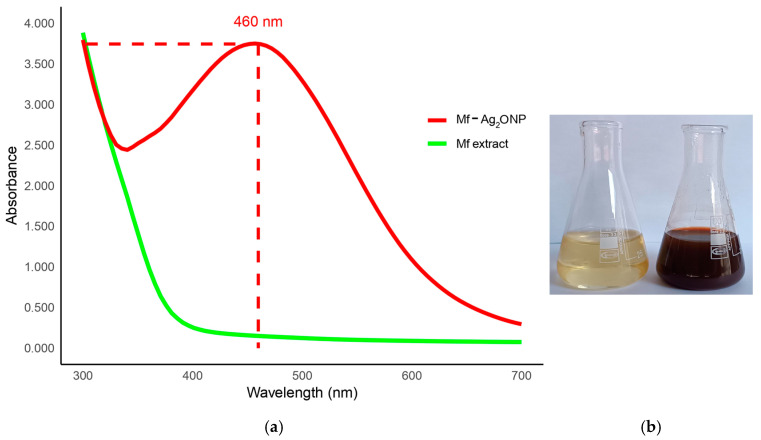
(**a**) UV-Vis spectrum of the nanoparticle dispersion and (**b**) color change in the synthesis process.

**Figure 2 nanomaterials-14-01875-f002:**
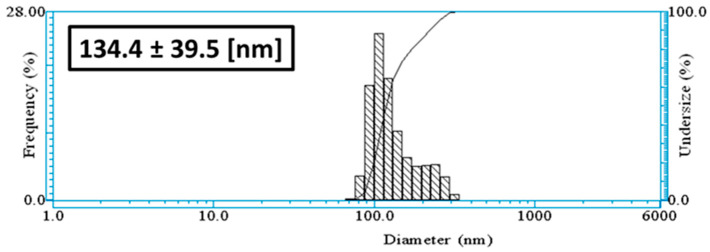
DLS: hydrodynamic diameter of Mf-Ag_2_ONPs.

**Figure 3 nanomaterials-14-01875-f003:**
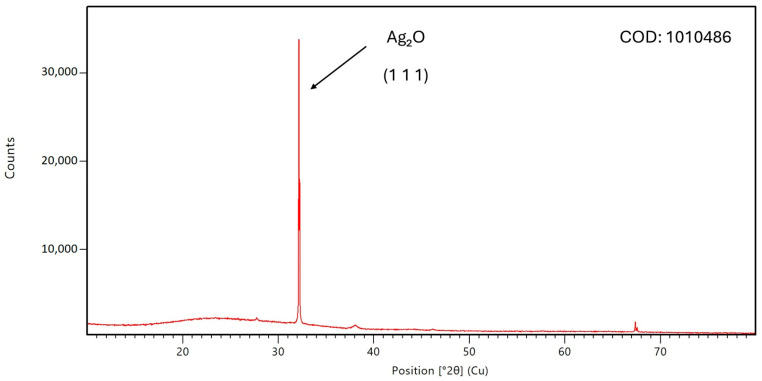
XRD diffractogram of Mf-Ag_2_ONPs.

**Figure 4 nanomaterials-14-01875-f004:**
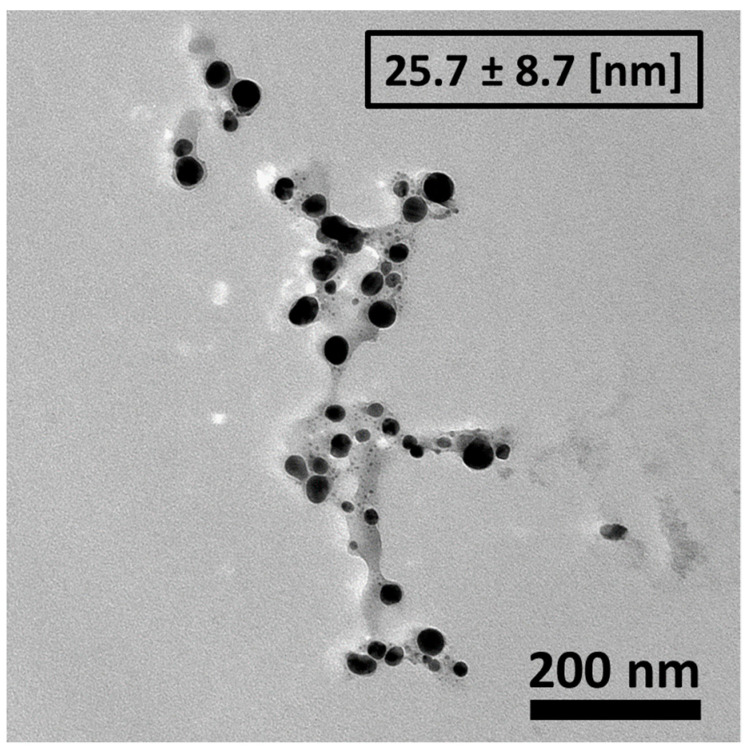
TEM image of synthesized Mf-Ag_2_ONPs.

**Figure 5 nanomaterials-14-01875-f005:**
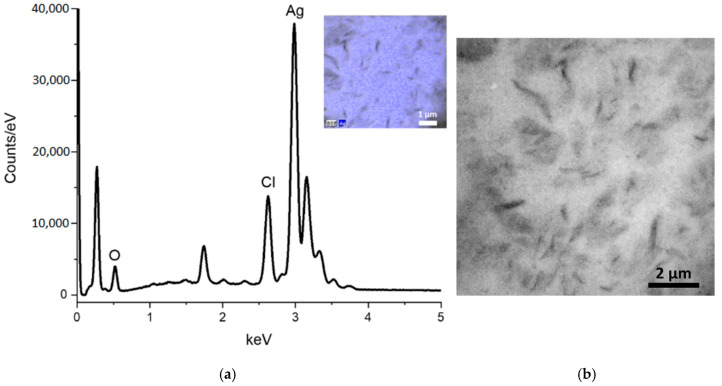
(**a**) EDS analysis and (**b**) SEM analysis of Mf-Ag_2_ONPs.

**Figure 6 nanomaterials-14-01875-f006:**
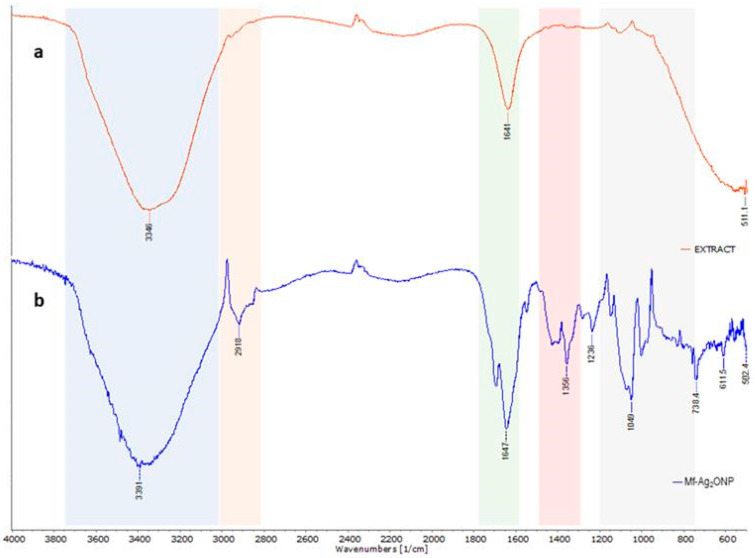
FTIR spectra of (**a**) extract of *M. flexuosa* and (**b**) Mf-Ag_2_ONPs.

**Figure 7 nanomaterials-14-01875-f007:**
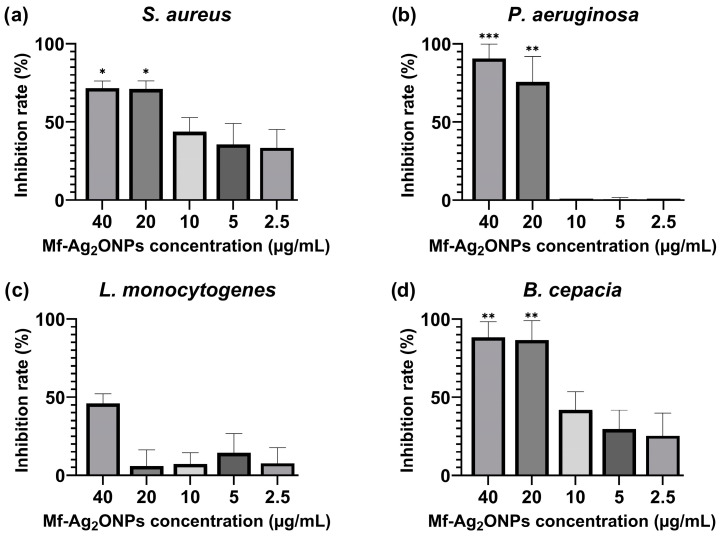
Percentage of biofilm inhibition of (**a**) *S. aureus* ATCC 25923, (**b**) *P. aeruginosa* ATCC 9027, (**c**) *L. monocytogenes* ATCC 13932, and (**d**) *B. cepacia* ATCC 25416 after 24 h incubation with Mf-Ag_2_ONP at a 2.5–40—µg/mL concentration. Treatments at different concentrations were compared with a 50% theoretical inhibition control for statistical significance using a two-way ANOVA test. All the values are mean ± SD, *p*-value (*) < 0.05, (**) < 0.01, and (***) < 0.001.

**Figure 8 nanomaterials-14-01875-f008:**
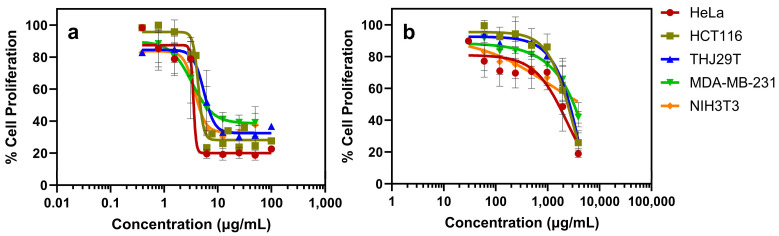
Dose–response curves of Mf-Ag_2_ONPs (**a**) and Mf extract (**b**) against tumor and non-tumor cell lines after 72 h incubation.

**Table 1 nanomaterials-14-01875-t001:** Phenolic compounds of aqueous extract of *M. flexuosa*.

	Concentration (mg/100 g DW)
Gallic acid	16.5 ± 0.6
Protocatechuic acid	538.1 ± 20.7
Syringic acid	28.1 ± 1.2
*m*-Coumaric acid	799.2 ± 74.8
Naringenin	155.7 ± 3.9
*p*-Cumaric acid	71.6 ± 0.2

**Table 2 nanomaterials-14-01875-t002:** Minimal inhibitory concentration (MIC) for different bacterial strains.

Bacterial Strain	MIC (µg/mL)
*Escherichia coli* ATCC 25922	22.5
*Staphylococcus aureus* ATCC 25923	22.5
*Pseudomonas aeruginosa* ATCC 27853	11.25
*Burkholderia cepacia* ATCC 25416	11.25
*Klebsiella pneumoniae* *	22.5
*Escherichia coli **	22.5
*Enterococcus faecium* *	45.0
*Salmonella enterica* serovar Kentucky *	22.5
*Pseudomonas aeruginosa* *	22.5

* MDR bacteria.

**Table 3 nanomaterials-14-01875-t003:** Tested strains and their corresponding MIC values.

Strain	MIC (µg/mL)
*Candida krusei* ATCC 14243	11.25
*Candida albicans* ATCC 10231	11.25
*Candida glabrata* ATCC 66032	5.63
*Candida tropicalis* ATCC 13803	>90

**Table 4 nanomaterials-14-01875-t004:** Antioxidant activity values for synthesized nanoparticles, fruit extract, and ascorbic acid after DPPH and ABTS/TEAC assays analysis.

Compound	DPPH IC_50_ (µg/mL)	ABTS IC_50_ (µg/mL)	TEAC * (µmol TE/g)
Mf-Ag_2_ONPs	14.77 ± 3.46	138.50 ± 38.52	3468.67 ± 419.82
Mf extract	227.40 ± 72.67	3596.67 ± 1314.00	123.16 ± 6.83
Ascorbic acid	5.01 ± 0.55	56.70 ± 6.76	4641.45 ± 483.93

* TEAC based on the ABTS assay only.

**Table 5 nanomaterials-14-01875-t005:** Half-maximal inhibitory concentration values (IC_50_) against tumor and non-tumor cell lines at 72 h (left) and therapeutic indexes (TI) (right). Cells only with media were used as negative control, and cisplatin (CDDP) was used as positive control.

IC_50_ (µg/mL)						TI			
Compound	HeLa	HCT116	THJ29T	MDAMB231	NIH3T3	HeLa	HCT116	THJ29T	MDAMB231
Mf-Ag_2_ONPs	3.5 ± 0.7	3.3 ± 0.4	5.5 ± 0.8	10.8 ± 2.4	3.7 ± 0.7	1.1	1.1	0.7	0.3
Mf extract	1336 ± 0.7	2307 ± 0.5	2400 ± 0.7	3691 ± 2.2	4048 ± 3.2	3.0	1.8	1.7	1.1
CDDP	2.3 ± 0.3	7.5 ± 0.1	10.6 ± 0.1	9.5 ± 0.1	3.7 ± 0.1	1.6	0.5	0.3	0.4

**Table 6 nanomaterials-14-01875-t006:** Hemolytic activity (%) of Mf-Ag_2_ONPs.

	% Hemolytic Activity
C−	0 ± 0.3
C+	100.0 ± 1.4
10 µg/mL	0.2 ± 0.1
20 µg/mL	1.7 ± 0.1
40 µg/mL	3.9 ± 0.4

C−—negative control (PBS 1x) and C+—positive control (10% Triton X-100).

## Data Availability

All data generated or analyzed during this study are included in this published article.

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
