# Peer review of "Green Synthesis of Silver Oxide Nanoparticles from Mauritia flexuosa Fruit Extract: Characterization and Bioactivity Assessment"

_nanomaterials, 2024, doi:10.3390/nano14231875_

Round 1

Reviewer 1 Report (Previous Reviewer 2)

Comments and Suggestions for Authors

Dear Authors,

I have read the new version (nanomaterials-3277367) of the article „Green Synthesis of Silver Oxide Nanoparticles from Mauritia flexuosa Fruit Extract: Characterization and Bioactivity Assessment”and I agree with the publication in the revised form. Please pay attention to some errors:

- Please note that Fig.1 has overlapped the legend.

- Check lines 958, 959.

Author Response

I have read the new version (nanomaterials-3277367) of the article „Green Synthesis of Silver Oxide Nanoparticles from Mauritia flexuosa Fruit Extract: Characterization and Bioactivity Assessment” and I agree with the publication in the revised form. Please pay attention to some errors:

- Please note that Fig.1 has overlapped the legend.

Thanks for the comment, we have corrected it.

- Check lines 958, 959.

Thanks for the comment, we have corrected it.

Reviewer 2 Report (Previous Reviewer 1)

Comments and Suggestions for Authors

The authors have partially addressed the issues in my comments. Zeta-potential values were added and some errors were corrected. Certainly, I would appreciate it if the authors could address the following question, as it represents the primary concern that is currently preventing my approval for the publication of this manuscript.

Table 1 showed the phenolic compounds of aqueous extract of M. flexuosa, are they the main contents in the extract? Which kind of compounds in the M. flexuosa reacted with the silver nitrate to give the Mf-AgONPs? How does the components of the fruit benefit the biomedical applications?

Author Response

The authors have partially addressed the issues in my comments. Zeta-potential values were added and some errors were corrected. Certainly, I would appreciate it if the authors could address the following question, as it represents the primary concern that is currently preventing my approval for the publication of this manuscript.

Table 1 showed the phenolic compounds of aqueous extract of M. flexuosa, are they the main contents in the extract? Which kind of compounds in the M. flexuosa reacted with the silver nitrate to give the Mf-AgONPs? How does the components of the fruit benefit the biomedical applications?

We appreciate the reviewer's observation and concern. Table 1 shows the main phenolic compounds identified in the freeze-dried aqueous extract of Mauritia flexuosa, which are consistent with those reported in preliminary studies by our research group (1)

In this context, the phenolic compounds found in this study showed a remarkable ability to reduce silver ions, due to their potential as electron donors (2). Among the most abundant compounds, naringenin  (3) and protocateic acid has been identified as an effective agent in the reduction of silver ions, as demonstrated in previous studies (4). Similarly, gallic acid is characterized by its high reactivity with silver ions, facilitating the rapid formation of nanoparticles (5,6). In the case of p-coumaric acid, a by-product of the oxidation of p-hydroxybenzoic acid, it contributes to both the stability and the formation of nanoparticles, as demonstrated in recent studies (7).

With the above and the results reported in this study, a further step is taken in the scientific field towards understanding the interaction between phenolic compounds and silver particles, although there is still a considerable gap for further study.

Bibliography

  1. Coyago-Cruz E, Guachamin A, Villacís M, Rivera J, Neto M, Méndez G, et al. Evaluation of bioactive compounds and antioxidant activity in 51 minor tropical fruits of ecuador. Foods. 2023 Dec 11;12(24):4439.
  2. Siddiqi KS, Husen A, Rao RAK. A review on biosynthesis of silver nanoparticles and their biocidal properties. J Nanobiotechnology. 2018 Feb 16;16(1):14.
  3. Ahmad A, Prakash R, Khan MS, Altwaijry N, Asghar MN, Raza SS, et al. Enhanced Antioxidant Effects of Naringenin Nanoparticles Synthesized using the High-Energy Ball Milling Method. ACS Omega. 2022 Sep 27;7(38):34476–84.
  4. Bhutto AA, Kalay Åž, Sherazi STH, Culha M. Quantitative structure–activity relationship between antioxidant capacity of phenolic compounds and the plasmonic properties of silver nanoparticles. Talanta. 2018 Nov;189:174–81.
  5. Al-Zahrani S, Astudillo-Calderón S, Pintos B, Pérez-Urria E, Manzanera JA, Martín L, et al. Role of Synthetic Plant Extracts on the Production of Silver-Derived Nanoparticles. Plants. 2021 Aug 13;10(8).
  6. Åžeker Karatoprak G, Aydin G, Altinsoy B, Altinkaynak C, KoÅŸar M, Ocsoy I. The Effect of Pelargonium endlicherianum Fenzl. root extracts on formation of nanoparticles and their antimicrobial activities. Enzyme Microb Technol. 2017 Feb;97:21–6.
  7. Guo J, Su K, Wang L, Feng B, You X, Deng M, et al. Poly(p-coumaric acid) nanoparticles alleviate temporomandibular joint osteoarthritis by inhibiting chondrocyte ferroptosis. Bioact Mater. 2024 Oct;40:212–26.

Reviewer 3 Report (New Reviewer)

Comments and Suggestions for Authors

The research manuscript by Zúñiga-Miranda et al. is reporting the biosynthesis of silver oxide nanoparticles with various biological activities. The manuscript can be considered for publication on nanomaterials after the following minor comments have been addressed:

1.      In the abstract, the abbreviation ‘’MDR’’ supposed to be written before ‘’pathogens’’ because it is an abbreviation of ‘’multidrug-resistant’’ not ‘’multidrug-resistant pathogens’’.

2.      The characterization techniques that confirmed the successful preparation of AgO nanoparticles should be mentioned in abstract.

3.      If the ‘’Nanoparticles’’ are abbreviated as ‘’NPs’’ in line 62, the abbreviation should be used throughout the manuscript instead of full name. Same apply in case of MDR and Ag2ONPs.

4.      Under sub-section 2.4., the purpose of using FTIR and EDS should be specified. UV-visible spectroscopy should be also mentioned under this sub-section since it was used as one of the techniques to confirm the formation of Mf-Ag2ONPs.

5.      This phrase ‘’HCT116 human colorectal carcinoma (human breast adenocarcinoma, ATCC No. CCL-247)’’ under sub-section 2.9. must be checked and confirmed. How come authors mentioned human colorectal carcinoma outside the bracket but mention human breast adenocarcinoma inside the brackets?

6.      It would be great if sub-section 2.9. and 3.8. can be renamed as ‘’cytotoxicity analysis’’ instead of ‘’anticancer activity’’ because authors don’t only evaluate cytotoxicity of Mf-Ag2ONPs against cancer cell lines but also against normal cells (NIH3T3 cells). 

Author Response

The research manuscript by Zúñiga-Miranda et al. is reporting the biosynthesis of silver oxide nanoparticles with various biological activities. The manuscript can be considered for publication on nanomaterials after the following minor comments have been addressed:

  1. In the abstract, the abbreviation ‘’MDR’’ supposed to be written before ‘’pathogens’’ because it is an abbreviation of ‘’multidrug-resistant’’ not ‘’multidrug-resistant pathogens’’.

Thanks for your comments. The abbreviation was changed to multidrug-resistant  (MDR) pathogens

  1. The characterization techniques that confirmed the successful preparation of AgO nanoparticles should be mentioned in abstract.

Thanks for your comments. The characterization techniques were mentioned in abstract

  1. If the ‘’Nanoparticles’’ are abbreviated as ‘’NPs’’ in line 62, the abbreviation should be used throughout the manuscript instead of full name. Same apply in case of MDR and Ag2ONPs.

Thanks for your comments. We have removed the abbreviation NPs because it is repeated many times in the document. We have maintained the abbreviations MDR and Ag2ONPs and have changed them in the document.

  1. Under sub-section 2.4., the purpose of using FTIR and EDS should be specified. UV-visible spectroscopy should be also mentioned under this sub-section since it was used as one of the techniques to confirm the formation of Mf-Ag2ONPs.

Thanks for your comments. We have mentioned Uv-visible under sub-section and the purpose of using FTIR and EDS for nanoparticle characterization was specified, as suggested by the reviewer.

  1. This phrase ‘’HCT116 human colorectal carcinoma (human breast adenocarcinoma, ATCC No. CCL-247)’’ under sub-section 2.9. must be checked and confirmed. How come authors mentioned human colorectal carcinoma outside the bracket but mention human breast adenocarcinoma inside the brackets?

Thank you for your observation; we have corrected the phrasing to consistently refer to HCT116 as "human colorectal carcinoma" and clarified that it corresponds to ATCC No. CCL-247.

  1. It would be great if sub-section 2.9. and 3.8. can be renamed as ‘’cytotoxicity analysis’’ instead of ‘’anticancer activity’’ because authors don’t only evaluate cytotoxicity of Mf-Ag2ONPs against cancer cell lines but also against normal cells (NIH3T3 cells). 

Thank you for your suggestion. While the focus of our study is on cancer, we believe that assessing the efficacy and specificity of Mf-Ag2ONPs requires evaluating their effects on both cancerous and non-cancerous cells (used as controls). Therefore, we prefer to retain the title "Anticancer Activity" for sub-sections 2.9 and 3.8, as it accurately reflects our emphasis on the compounds' activity in cancer contexts while acknowledging the role of normal cells in demonstrating selective toxicity.

Round 2

Reviewer 2 Report (Previous Reviewer 1)

Comments and Suggestions for Authors

I remain unconvinced due to the lack of a good reason demonstrating the superiority of using M. flexuosa extract for the preparation of AgONPs as a bactericide over other reductants. However, I acknowledge the authors' efforts to enhance the quality of their paper. I am inclined to approve the publication of the manuscript, with the hope that the authors will address this concern in the final version for publication.

Author Response

We appreciate the reviewer’s insightful question regarding the comparative advantage of using Mauritia flexuosa extract in the synthesis of Agâ‚‚ONPs. While demonstrating the superiority of this extract over other reductants was not the primary objective of our study, we emphasize its potential as an alternative to hazardous chemicals due to its high content of bioactive compounds, which facilitate an environmentally friendly synthesis of nanoparticles in alignment with several principles of green chemistry.

This study seeks to contribute to the growing field of green nanoparticle synthesis using tropical fruit extracts, particularly because most research has focused on plant species traditionally recognized for their high phenolic content. Our findings indicate that M. flexuosa provides a viable, human- and eco-friendly solvent system for nanoparticle production, as its bioactive profile not only effectively reduces silver ions but also enhances biocompatibility, making it suitable for biomedical applications. This work thus supports the potential of M. flexuosa as a valuable addition to sustainable nanoparticle synthesis

This manuscript is a resubmission of an earlier submission. The following is a list of the peer review reports and author responses from that submission.

Round 1

Reviewer 1 Report

Comments and Suggestions for Authors

In this manuscript, Mauritia flexuosa fruit extract was used to prepare silver oxide nanoparticles and their biological activities were assessed. The results demonstrated that the silver oxide nanoparticles could be used as antimicrobial against bacterial, fungal and biofilm. The nanoparticles with fruit extract also exhibited antioxidant and anticancer effects. However, the characterization of the material is very preliminary, and the results are shown in very poor from. At present, I cannot recommend publication of this manuscript at any scientific journals. Some major issues are listed as follows:

1.        Table 1 showed the phenolic compounds of aqueous extract of M. flexuosa, are they the main contents in the extract? Which kind of compounds in the M. flexuosa reacted with the silver nitrate to give the Mf-AgONPs? How does the components of the fruit benefit the biomedical applications?

2.        The unit of the scale bar in the TEM image should be nm rather than μm.

3.        Some characterization data are lacking, such as the zeta potential value, the ratio of the AgONPs to the organic compounds in the Mf-AgONPs.

4.        The hydrated size is much larger than the nanoparticle observed under TEM, why?

5.        The quality of the SEM image is very poor. What is the needle-like structure observed in SEM?

6.        Please revise the “anti-tumor” as “anti-cancer” as only cellular studies are performed in this work. In addition, what is the toxicity of the nanoparticles to normal cell lines?

Comments on the Quality of English Language

The English should be improved.

Reviewer 2 Report

Comments and Suggestions for Authors

In this research, a green method was used in order to synthesize silver oxide nanoparticles using Mauritia flexuosa Fruit Extract as reducing, stabilizing and coating agent.

The total polyphenol content from the extract was determined. The characterization of the obtained silver oxide nanoparticles is supported by the Dynamic Light Scattering (DLS), X-ray diffraction (XRD), Transmission Electron Microscopy (TEM), Energy-dispersive X-ray spectroscopy (EDS), Scanning Electron Microscopy (SEM), UV-visible spectroscopy and FTIR analysis. Their antibacterial and antifungal activities were evaluated against non-resistant bacterial strains as well as multidrug-resistant bacterial strains and Candida species. The hemolytic activity was also studied. Moreover, the biofilm inhibitory properties, antioxidant capacities and anti-tumor effects against various cancer cell lines were investigated. The results are accompanied by 6 tables and 8 figures. Supplementary material of the current manuscript contains the percentage of biofilm inhibition activity and data from antitumor activity of biosynthesized silver oxide nanoparticles and Mauritia flexuosa fruit extract. The authors highlight the importance of their own results by comparing them with those reported by other studies using plant extracts for nanoparticle synthesis, thus suggesting potential clinical applications.

I agree with the publication of the „Green Synthesis of Silver Oxide Nanoparticles from Mauritia flexuosa Fruit Extract: Characterization and Bioactivity Assessment” paper in Nanomaterials journal after minor revision and clarification of the aspects mentioned as following:

- It is not clear whether you obtained AgO or Ag2O. According to the X-ray diffractogram and card no. 1010488 (see Fig. 3), you obtained Ag2O. AgO is mentioned in the text (lines 25, 27 etc., see everywhere in the text at the mention of Mf-AgONPs). Or is AgO a sample code and not the oxide formula? For better understanding, you should specify this.

- Reference [29] (line 360) is related to Ag nanoparticles. Please change it.